# Clinical Risk-Aware Multi-Level Grading for Coronary Artery Stenosis through Curved Feature Reconstruction

**Shishuang Zhao**[*1] 🆔                                   SHISHUANG.ZHAO@YIZHUN-AI.COM
**Hongtai Li**[*2]                                               20201099@CMU.EDU.CN
**Junjie Hou**[1]                                        JUNJIE.HOU@YIZHUN-AI.COM
**Yuhang Liu**[†1] 🆔                                    YUHANG.LIU@YIZHUN-AI.COM
[1] *Yizhun Medical AI Co., Ltd*

[2] *The First Hospital of China Medical University*

**Editors:** Accepted for publication at MIDL 2026

## Abstract

Developing a multi-level grading model for coronary artery stenosis holds great clinical significance for the diagnosis of coronary artery disease. However, designing an effective multi-level deep learning algorithm faces significant challenges. Specifically, utilizing CCTA or 3D SCPR images alone presents inherent shortcomings: CCTA images are difficult to analyze due to the tortuous paths of blood vessels, while 3D SCPR images are prone to abnormal distortions that hinder accurate grading. Furthermore, different stenosis grades are associated with varying clinical risks, and incorporating this association into the algorithm is non-trivial. To address the former problems, we propose the Curved Feature Reconstruction (CFR) module, which uses vessel curves as prior and employs a point-by-point correspondence strategy to precisely align and fuse features from both 3D SCPR and CCTA images. Meanwhile, a Clinical Risk-Aware (CR) Loss is employed to introduce clinical risk relevance into the network training so that the algorithm can better align with the clinical diagnosis. The experimental results on a in-house dataset reveal that our approach significantly outperforms other methods, and several ablation studies also demonstrate the effectiveness of our proposed designs.

**Keywords:** Multi-level stenosis grading, Deep learning, CCTA, 3D SCPR

## 1. Introduction

Coronary artery stenosis, which refers to the narrowing of coronary arteries, has been recognized as the primary indicator of coronary artery disease (CAD)(Jensen et al., 2020; Naghavi et al., 2003; Otsuka et al., 2013; Hou et al., 2024; Wang et al., 2022; Zhang et al., 2023; Zhao et al., 2025). Accurate grading of stenosis is crucial for the effective diagnosis of CAD through Coronary Computed Tomographic Angiography (CCTA), a widely used non-invasive diagnostic examination(Mowatt et al., 2008). In clinical practice, stenosis can be graded on a 5-level scale, from 1 to 5, representing the increasing degrees of stenosis: minimal (1–24%), mild (25–49%), moderate (50–69%), severe stenosis (70–99%), and occluded vessel (100%)(Cury et al., 2022). Different stenosis grades correspond to different heart health risks, which leads to various medical interventions and treatments. Therefore,

---

[*] Contributed equally

[†] Corresponding Author

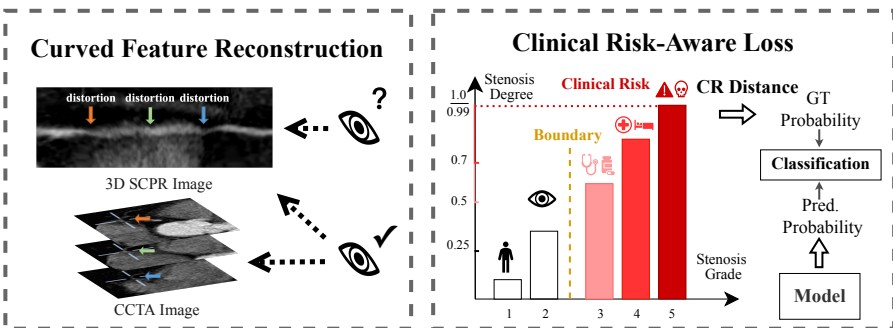

Figure 1: Illustrations of the challenges and our solutions. In the left figure, identifying suspicious stenosis solely through 3D SCPR images is difficult due to abnormal distortions (indicated by colored arrows), while inclusion of CCTA images can provide more comprehensive information for stenosis grading. In the right figure, we propose CR Loss to encode the important clinical risk boundary explicitly into the model training.

achieving accurate multi-level stenosis grading is a significant step towards the development of automatic computer-aided diagnosis system for CAD.

To date, several studies have focused on multi-level stenosis grading(Zreik et al., 2018; Ma et al., 2021; Tejero-de Pablos et al., 2019; Kirişli et al., 2013; Kelm et al., 2011; Zhang et al., 2022). However, a common issue with these works is the insufficient incorporation of prior clinical knowledge, as shown in Figure 1. **Firstly**, clinicians primarily analyze CCTA images to diagnose CAD, while previous studies(Zreik et al., 2018; Ma et al., 2021; Zhang et al., 2022) have relied on 3D Straightened Curved Planar Reformation (3D SCPR) images, which may impede the analysis process due to abnormal distortions caused by the reformation mechanism, although 3D SCPR images provide information on the 3D geometry of arteries. **Secondly**, clinicians have different medical treatments for different stenosis grades, and the observation of moderate or greater stenosis serves as a critical diagnostic criterion for CAD(Cury et al., 2022). As the primary expert consensus document, CAD-RADS 2.0(Cury et al., 2022) provides a standardized framework for grading stenosis from 1 to 5. Crucially, it also states that patients with stenosis grade greater than 2 are at higher risk and may require further assessments and anti-anginal therapy, while those with grade 1 or 2 only need preventive pharmacotherapy(Cury et al., 2022). Thus, a clinical risk boundary exists between grade 1/2 and grade 3/4/5, and misclassification on either side of the boundary can have serious consequences. However, previous studies did not consider the clinical risk relevance of stenosis grading. They either focused on crude binary classification(Zreik et al., 2018; Ma et al., 2021; Tejero-de Pablos et al., 2019) or stenosis rate regression(Kirişli et al., 2013; Kelm et al., 2011; Zhang et al., 2022), treating different stenosis grades equally, which does not meet the demands of clinical practice. Additionally, accurately defining and acquiring stenosis rates is challenging due to the complex nature of the coronary lumen space.

To address the shortcomings of previous methods, we propose a novel deep learning-based approach for achieving multi-level stenosis grading. Our approach consists of two key designs to tackle the aforementioned problems. Firstly, we introduce the *Curved Feature Reconstruction* (CFR) module to address the limitations of 3D SCPR images. The CFR module integrates features from CCTA images with features from 3D SCPR images, using vessel curve priors to establish a connection between the two image types. By combining the undistorted CCTA images, which retain the original texture details around stenosis regions, with 3D SCPR images, a more comprehensive analysis of stenosis can be achieved. Secondly, to align network training with clinical practice, we propose a novel loss function called *Clinical Risk-Aware Loss* (CR Loss). Specifically, we define the CR Distance to quantify the dissimilarity between different grades. We treat stenosis grading as a classification task, with the CR Distance serving as the metric function for the encoding of the ground truth probability distribution, explicitly incorporating clinical relevance into the network.

Our contributions are three-fold. (1) We propose the CFR module, which enables comprehensive stenosis analysis from both 3D SCPR and CCTA images. (2) We introduce the CR Loss, a novel loss function that explicitly encodes the important clinical risk boundary into network training. (3) We introduce two new metrics, MCRE and AMCRE, which measure the clinical risk relevance of stenosis grading results. Our method significantly outperforms other approaches in these metrics, indicating its better alignment with clinical needs.

## 2. Related Works

Regarding model input, most previous studies solely use 3D SCPR images as input(Ma et al., 2025, 2024, 2021; Zreik et al., 2018; Zhang et al., 2022; Candemir et al., 2020), which are prone to abnormal distortions and lacks the original texture details. In terms of methodology, previous methods for stenosis grading can be divided into two categories: stenosis rate regression(Kirişli et al., 2013; Kelm et al., 2011; Shahzad et al., 2013; Zhang et al., 2022) and binary classification(Zreik et al., 2018; Ma et al., 2021; Denzinger et al., 2019; Tejero-de Pablos et al., 2019). However, estimating vessel lumen accurately for stenosis rate calculation is a challenging task due to the complex nature of coronary lumen space, and the stenosis grade cannot be determined if the reference lumen is either non-existent or unreliable(Cury et al., 2022). Moreover, crude binary classification is not sufficient to meet the clinical requirements. Therefore, none of these approaches fulfills the requirements of the multi-level grading mechanism in clinical diagnosis. In this paper, we present the first deep learning-based multi-level stenosis grading method to facilitate the practical utilization.

## 3. Method

In this section, we will elaborate on the details of our method. Let $\mathbf{I} \in \mathbb{R}^{H \times W \times D}$, $\mathcal{S} = \{\mathbf{p}_i\}_{i=1}^{N}$ and $\overline{\mathcal{S}} = \{\mathbf{p}_i | \mathbf{p}_i \in \mathcal{S}\}_{i=s}^{e}$ denote the CCTA image, the centerline points, and stenosis instance respectively, where $\mathbf{p}_i \in \mathbb{R}^3$ is the 3D point location; $N$ is the number of centerline points; and $1 \leq s < e \leq N$. The stenosis instance is a vessel segment consisting of several consecutive centerline points actually. Given a CCTA image and a stenosis instance $\overline{\mathcal{S}}$, the goal of the stenosis grading task is to predict the grade $y \in \{1, 2, 3, 4, 5\}$ for $\overline{\mathcal{S}}$.

Our system pipeline (Figure 2) can be divided into two parts: 1) the *Curved Feature Reconstruction* module to extract features from 3D SCPR image and CCTA patch and fuse them to predict the stenosis grade (Section 3.2); and 2) a novel *Clinical Risk-Aware Loss* to incorporate clinical relevance into the network training (Section 3.3). The following sections provide details of our method.

### 3.1. Generation Methodology of 3D SCPR Images

Given a CCTA image $\mathbf{I}$ and a sequence of uniformly sampled centerline coordinates $\mathcal{S} = \{\mathbf{p}_i\}_{i=1}^{N}$ (which are pre-computed by external algorithms and are beyond the scope of this study), a 3D SCPR image can be generated(Kanitsar et al., 2002).

For each $i \in [1, N]$, we compute the unit tangent vector $\mathbf{T}_i$. Subsequently, we derive two mutually orthogonal unit vectors, the normal $M_i$ and binormal $\mathbf{B}_i$, which define the cross-sectional plane perpendicular to the centerline path at $p_i$. On each cross-sectional plane, $H_\text{s} \times W_\text{s}$ coordinates are sampled to form a planar grid, with $p_i$ positioned precisely at the grid's center. By performing bilinear interpolation on these coordinates within the CCTA image $\mathbf{I}$, a 2D cross-sectional slice is obtained. After repeating this process for all $N$ points, the resulting slices are stacked sequentially to construct a 3D SCPR image $\mathbf{I}_\text{scpr} \in \mathbb{R}^{H_\text{s} \times W_\text{s} \times N}$. The code used to generate the 3D SCPR images is publicly available at https://github.com/zhaoshishuang/3D-SCPR-Generation.

By incorporating the vessel centerline as prior information, 3D SCPR images enable the algorithm to perceive the vessels in a straightened state, thereby simplifying the identification of stenosis. However, due to the inherent torsion of 3D centerlines, local planar grids tend to rotate inconsistently at different positions. This leads to rotational misalignment, where the radial orientation of points fails to align, causing anatomical structures to appear twisted or spiraled in the final 3D SCPR images. This is a systemic issue inherent to the design.

### 3.2. Curved Feature Reconstruction

Previous works(Ma et al., 2021; Zreik et al., 2018; Zhang et al., 2022; Candemir et al., 2020) usually overlook the presence of abnormal distortions in 3D SCPR images, which may adversely affect stenosis grading. In this work, we incorporate the undistorted CCTA images into the network inputs to provide original texture around stenosis regions that may be lost in 3D SCPR images.

Firstly, as the stenosis grade is determined by comparing the stenotic part with the normal part, we extend stenosis instance $\overline{\mathcal{S}} = \{\mathbf{p}_i | \mathbf{p}_i \in \mathcal{S}\}_{i=s}^{e}$ along the vessel centerline to get $\hat{\mathcal{S}} = \{\mathbf{p}_i | \mathbf{p}_i \in \mathcal{S}\}_{i=s-m}^{e+m}$, where $m$ is a hyper-parameter, to increase the vascular perceptual field. Then the 3D SCPR image $\mathbf{I}_\text{scpr} \in \mathbb{R}^{H_\text{s} \times W_\text{s} \times D_\text{s}}$ is obtained given the centerline points $\hat{\mathcal{S}}$ and the original CCTA image $\mathbf{I}$. Besides, we crop a patch $\mathbf{I}_\text{ccta} \in \mathbb{R}^{H_\text{c} \times W_\text{c} \times D_\text{c}}$ from CCTA image $\mathbf{I}$, which fully contains the vascular regions of $\hat{\mathcal{S}}$, to preserve more texture details. Finally, we feed $\mathbf{I}_\text{scpr}$ and $\mathbf{I}_\text{ccta}$ into the network to extract features separately and fuse them together to predict the stenosis grade.

**Feature Extraction and Fusion.** We utilize two separate 3D ResUNet(Yu et al., 2017) as backbones to extract 3D SCPR features $\mathbf{F}_\text{scpr} \in \mathbb{R}^{H_\text{s} \times W_\text{s} \times D_\text{s} \times C}$ and CCTA features $\mathbf{F}_\text{ccta} \in \mathbb{R}^{H_\text{c} \times W_\text{c} \times D_\text{c} \times C}$, where $C$ is the channel size.

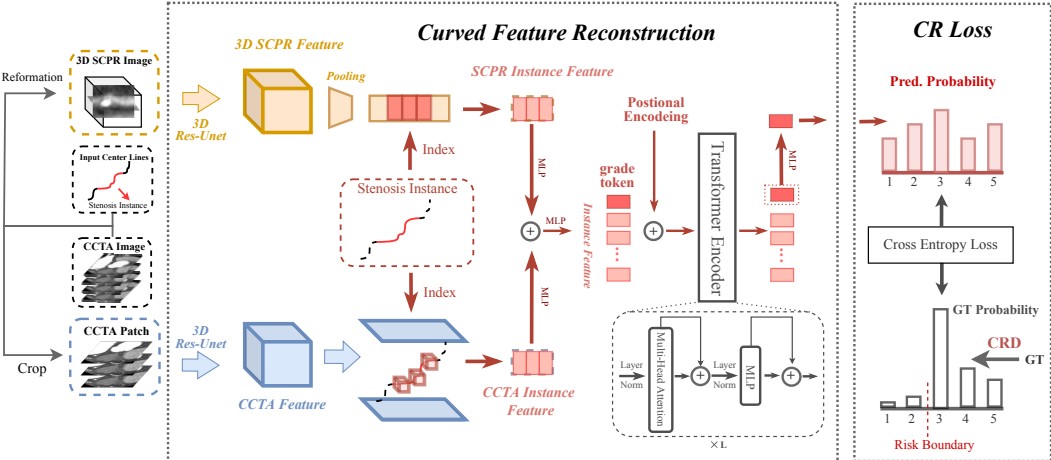

Figure 2: The overall framework of our method. Firstly, We obtain CCTA patch and 3D SCPR image as model inputs using CCTA image and input centerlines. Then CFR module is adopted to extract and fuse features from the inputs for stenosis grading. To incorporate clinical risk boundaries into neural network training, we finally apply a novel CR Loss for loss calculation.

As $\mathbf{F}_{\mathrm{ccta}}$ has the same shape as $\mathbf{I}_{\mathrm{ccta}}$, the CCTA stenosis instance features can be directly acquired by indexing the stenosis instance coordinates $\mathbf{p}_i \in \overline{\mathcal{S}}$ from $\mathbf{F}_{ccta}$. We then use an MLP to project the instance features into new space and obtain $\overline{\mathbf{F}}_{\mathrm{ccta}} \in \mathbb{R}^{(e-s+1)\times C}$.

$$\overline{\mathbf{F}}_{\mathrm{ccta}} = \mathrm{MLP}\left(\left[\mathbf{F}_{\mathrm{ccta}}[\mathbf{p}_s], ..., \mathbf{F}_{\mathrm{ccta}}[\mathbf{p}_e]\right]\right) \tag{1}$$

where $\mathbf{p}_i \in \overline{\mathcal{S}}, \mathbf{F}_{\mathrm{ccta}}[\mathbf{p}_i]$ means the local feature of $\mathbf{F}_{\mathrm{ccta}}$ at position $\mathbf{p}_i$.

To obtain the 3D SCPR stenosis instance features, we first apply a 2D average pooling on $\mathbf{F}_{\mathrm{scpr}}$ along the dimensions of $H_{\mathrm{s}}$ and $W_{\mathrm{s}}$ to get $\mathbf{F}'_{\mathrm{scpr}} \in \mathbb{R}^{D_{\mathrm{s}}\times C}$. Subsequently, the stenosis instance features are indexed from $\mathbf{F}'_{\mathrm{scpr}}$ using the position of stenosis instance $\overline{\mathcal{S}}$ in points set $\hat{\mathcal{S}}$. And an MLP is then employed to generate $\overline{\mathbf{F}}_{\mathrm{scpr}} \in \mathbb{R}^{(e-s+1)\times C}$.

$$\overline{\mathbf{F}}_{\mathrm{scpr}} = \mathrm{MLP}\left(\left[\mathbf{F}'_{\mathrm{scpr}}[m], ..., \mathbf{F}'_{\mathrm{scpr}}[m+e-s]\right]\right) \tag{2}$$

We then apply the element-wise addition on $\overline{\mathbf{F}}_{\mathrm{ccta}}$ and $\overline{\mathbf{F}}_{\mathrm{scpr}}$ to get the texture-enhanced stenosis instance features $\mathbf{F} = \overline{\mathbf{F}}_{\mathrm{ccta}} + \overline{\mathbf{F}}_{\mathrm{scpr}}$.

**Grade Prediction.** Transformer(Vaswani et al., 2017) has been proven to perform well in many computer vision tasks(Dosovitskiy et al., 2020; Liu et al., 2021). Here we also apply an attention-based network for stenosis grade prediction. Following (Dosovitskiy et al., 2020), we first add a learnable grade token $\mathbf{f}_{\mathrm{grade}}$ to the stenosis instance features $\mathbf{F} = [\mathbf{f}^1, ..., \mathbf{f}^{e-s+1}]$, where $\mathbf{f}^i \in \mathbb{R}^C$ is the $i$-th token of $\mathbf{F}$, to form a sequence of tokens $[\mathbf{f}_{\mathrm{grade}}, \mathbf{f}^1, ..., \mathbf{f}^{e-s+1}]$. Then the tokens are added by learnable position embeddings $\mathbf{E}_{pos} \in$

$\mathbb{R}^{(e-s+2)\times C}$ and the resulting sequence serves as the input of the Transformer encoder, which contains multiple layers of multi-head self-attention (MSA) and MLP blocks.

$$
\begin{aligned}
\mathbf{g}_0 &= [\mathbf{f}_{\text{grade}}, \mathbf{f}^1, ..., \mathbf{f}^{e-s+1}] + \mathbf{E}_{pos} \\
\mathbf{g}'_l &= \text{MSA}(\text{LN}(\mathbf{g}_{l-1})) + \mathbf{g}_{l-1}, \ l = 1 \ ... \ L \\
\mathbf{g}_l &= \text{MLP}(\text{LN}(\mathbf{g}'_l)) + \mathbf{g}'_l, \ l = 1 \ ... \ L \\
\mathbf{g} &= \text{LN}(\mathbf{g}^0_L)
\end{aligned}
\tag{3}
$$

where $L$ is the number of layers in the encoder. The resulting feature $\mathbf{g} \in \mathbb{R}^C$ is then fed into an MLP to predict the final grade probability $\hat{P}$.

$$
\hat{P} = \text{Softmax}(\text{MLP}(\mathbf{g})) \in \mathbb{R}^5
\tag{4}
$$

### 3.3. Clinical Risk-Aware Loss

Clinically, patients with different stenosis grades have varying clinical risks, requiring various treatment methods(Cury et al., 2022). Therefore, an effective stenosis grading metric must be clinically risk-aware, applying a large penalty when predicted risk significantly deviates from the ground truth risk.

**CR Distance.** Following the above intuition, we propose a novel Clinical Risk-Aware Distance (CR Distance) as the distance metric to measure the dissimilarity between prediction and ground truth. CR Distance takes into account the clinical risks associated with stenosis grades, which can be formulated as follows:

$$
\text{CRD}(a,b) = \begin{cases} (|a-b|+1)^2, & a \leq 2 < b \ \text{ or } \ b \leq 2 < a \\ |a-b|, & \text{else} \end{cases}
\tag{5}
$$

where a clinical risk boundary is between $1/2$ and $3/4/5$. When both inputs are on one side of the boundary, the CR Distance is equal to the average error as this type of mistake is not as significant. However, when inputs are on either side of the boundary, the CR Distance takes the form of a square error, with a penalty term 1 added. As a result, the distance between the grades on either side of the boundary measured by CR Distance is much larger, which is compatible with the clinical characteristic of stenosis grade.

**CR Loss.** Stenosis grading is an ordinal regression problem and can be handled naturally by K-rank ordinal regression. However, it is hard to leverage the clinical risk boundary for K-rank ordinal regression as it actually performs binary classification during training. In our work, we approach stenosis grading as a classification task with a soft ground truth target(Diaz and Marathe, 2019) $P = [P_1, ..., P_5]$:

$$
P_i = \frac{\exp(-\phi(gt,i))}{\sum_{j=1}^5 \exp(-\phi(gt,j))},
\tag{6}
$$

where $\phi(gt,i)$ is a metric function that measures the distance between the ground truth label $gt$ and grade $i \in \{1,2,3,4,5\}$.

In this work, we propose to use the CR Distance as the metric function $\phi$ to introduce the clinical characteristics of stenosis into model training. As a result, the soft ground truth

target can better reflect the clinically unacceptable confusion between stenosis grades $1/2$ and $3/4/5$. Our CR Loss is calculated by:

$$\text{CR Loss} = -w_{gt} \sum_{i=1}^{5} P_i \log(\hat{P}_i) = -w_{gt} \sum_{i=1}^{5} \frac{\exp(-\text{CRD}\,(gt, i))}{\sum_{j=1}^{5} \exp(-\text{CRD}\,(gt, j))} \log(\hat{P}_i) \qquad (7)$$

where $w_{gt} \in \mathbf{w}$ is a loss weight for $gt$ grade and $\mathbf{w} = [w_1, w_2, w_3, w_4, w_5]$ is a loss weights vector for all grades, as the number of each grade level is imbalanced. We also compare CR Distance with other metric functions (like absolute distance (AD) and square distance (SD)) in Table 4 to demonstrate its superiority.

## 4. Experiment

### 4.1. Experimental Settings

#### 4.1.1. DATASET.

We collect a private dataset comprised of 500 CCTA scans, which contains labels of centerline points of coronary arteries and stenosis instances with corresponding grades annotated by at least three experienced radiologists. The dataset is collected in compliance with the terms of the licensing agreement and ethical certification. The number of stenosis instances in each grade level from 1 to 5 is 1654, 1443, 511, 266, and 40 respectively. We randomly split the dataset into train, validation and test set with a ratio of 3:1:1.

**Evaluation metrics.** Stenosis grading is essentially an ordinal regression task, and traditional evaluation metrics such as Quadratic Weighted Kappa (**QWK**)(Warrens, 2012), Mean Absolute Error (**MAE**), and Average Mean Absolute Error (**AMAE**) are naturally suited for it. However, these metrics do not take into account the clinical risk associated with each stenosis grade. CR distance incorporated with clinical risk boundary is suitable for measuring the clinical risk relevance of stenosis grading results. Therefore, we introduce two new metrics based on CRD: Mean Clinical Risk-Aware Error (**MCRE**) and Average Clinical Risk-Aware Error (**AMCRE**), MCRE is defined as the average of CRD over all instances: $\text{MCRE} = \frac{1}{N} \sum_{i=1}^{N} \text{CRD}(pred_i, gt_i)$ and AMCRE is calculated as the average of MCRE for all classes $\text{AMCRE} = \frac{1}{C} \sum_{i=1}^{C} \text{MCRE}_i$.

**Implementation details.** The loss weights $\mathbf{w}$, extend length $m$, channel size $C$, number of transformer layers $L$ and attention head number are set to $[1, 1, 2, 3, 4]$, 10, 64, 8 and 4 by default. The proposed model is trained for 150 epochs with 15 epochs warm up and cosine learning rate decay schedule. AdamW(Loshchilov and Hutter, 2017) optimizer is adopted with learning rate of 0.001, weight decay of 0.001 and batch size of 32. All experiments are conducted on 8 NVIDIA GeForce RTX 3090 GPUs.

### 4.2. Experiment Results

In Table 1, we compare our method with other approaches, including Random Forest Regression(Kelm et al., 2011), Coronary R-CNN(Zhang et al., 2022) and Cost-sensitive Classification(Galdran et al., 2020). Several baselines based on hard label multi-class classification, Continuous Label Regression and K-rank ordinal regression(Niu et al., 2016) are also provided for comparison. The results show that our approach outperforms all other methods

Table 1: Comparison with other methods.

| Method | Data Type | QWK↑ | MAE↓ | AMAE↓ | MCRE↓ | AMCRE↓ |
|---|---|---|---|---|---|---|
| Random Forest Regression | 3D SCPR | 0.378 | 0.684 | 1.141 | 2.224 | 4.555 |
| Coronary R-CNN | 3D SCPR | 0.497 | 0.603 | 0.829 | 1.969 | 2.791 |
| Multi-class Classification | 3D SCPR | 0.611 | 0.574 | 0.744 | 1.761 | 2.156 |
| Cost-sensitive Classification | 3D SCPR | 0.673 | 0.515 | 0.598 | 1.503 | 1.575 |
| Continuous Label Regression | 3D SCPR | 0.583 | 0.597 | 0.733 | 1.901 | 2.117 |
| K-rank Ordinal Regression | 3D SCPR | 0.672 | **0.436** | 0.569 | 1.585 | 1.425 |
| **Ours** | CCTA+3D SCPR | **0.712** | 0.480 | **0.533** | **1.296** | **1.186** |

Table 2: Stratified comparison with other methods.

| Method | MAE↓ | | | | | MCRE↓ | | | | |
|---|---|---|---|---|---|---|---|---|---|---|
| | 1 | 2 | 3 | 4 | 5 | 1 | 2 | 3 | 4 | 5 |
| Random Forest Regression | 0.416 | 0.728 | 0.852 | 1.209 | 2.500 | 1.465 | 2.272 | 2.113 | 4.224 | 12.70 |
| Coronary R-CNN | 0.441 | 0.735 | 0.487 | 0.881 | 1.600 | 1.818 | 2.206 | 1.139 | 2.493 | 6.300 |
| Multi-class Classification | 0.419 | 0.699 | 0.513 | 0.791 | 1.300 | 1.524 | 2.125 | 1.217 | 2.015 | 3.900 |
| Cost-sensitive Classification | 0.352 | 0.662 | 0.487 | 0.687 | 0.800 | 1.225 | 1.967 | 1.165 | **1.418** | 2.100 |
| Continuous Label Regression | 0.432 | 0.689 | 0.591 | 0.955 | 1.000 | 1.600 | 2.125 | 1.470 | 3.090 | 2.300 |
| K-rank Ordinal Regression | **0.127** | 0.676 | 0.513 | 0.731 | 0.800 | 1.349 | 2.007 | 1.191 | 1.776 | 0.800 |
| **Ours** | 0.292 | **0.654** | **0.478** | **0.642** | **0.600** | **0.798** | **1.923** | **1.130** | 1.478 | **0.600** |
| Sample count per grade | 315 | 272 | 115 | 67 | 10 | 315 | 272 | 115 | 67 | 10 |

by a large margin on almost all metrics. Particularly, our method exhibits superior performance in the metrics of MCRE and AMCRE, which are relevant to distinguish stenosis grades between 1/2 and 3/4/5. This aligns with the design objective of CR loss indicating the clinical risk-aware ability has been encoded into the network through CR loss.

Table 2 provides a stratified performance analysis across individual grades, with the sampling count for each grade reported in the bottom row. We specifically report MAE and MCRE, as AMAE/AMCRE are equivalent to MAE/MCRE in this context, and QWK is mathematically inapplicable. As observed, our method achieves state-of-the-art performance in most grades, with the most significant gains observed in grade 5. Since grade 5 represents the highest clinical risk, misclassifying these cases as grade 1/2 would lead to delayed intervention and severe consequences. This stratified superiority underscores the clinical reliability of our method.

### 4.3. Ablation Study

**Curved Feature Reconstruction.** In Table 3, we ablate the effectiveness of different data utilization strategies in CFR module, including CCTA combined with 3D SCPR, CCTA only, and 3D SCPR only. The results demonstrate that both 3D SCPR images and CCTA images are critical for accurate stenosis grading, and the combination of them can achieve the best performance.

Furthermore, we qualitatively validate this conclusion through visualization. Specifically, we randomly select 10 samples from the test set and use Grad-CAM(Selvaraju et al., 2017) to generate activation heat maps of the last layer in the backbone network. These

heat maps are visualized in Figure 3, and to improve the visualization quality, we project and interpolate the generated 3D activation map into a 2D SCPR image. Since our method utilizes two separate backbones, we average the activation maps from these two backbones to generate a single combined activation map.

Our visualization results indicate that neither 3D SCPR images nor CCTA images can accurately identify the stenosis region on their own. However, the combination of them can effectively learn and highlight the accurate stenosis region. Therefore, our CFR module can improve performance by leveraging the complementary information from both 3D SCPR and CCTA images.

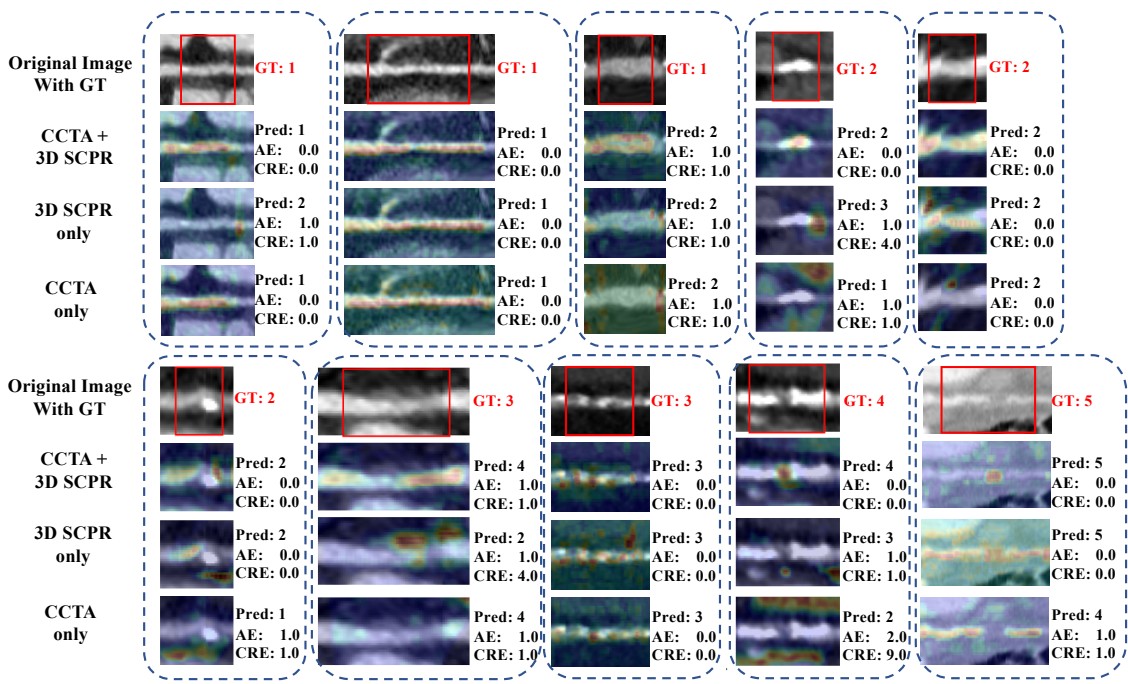

Figure 3: Visual comparison of results using different data utilization strategies. The text on the right displays the ground truth (GT) in red, while the predicted results (Pred) and evaluation metrics: Absolute Error (AE) and Clinical Risk-Aware Error (CRE), are shown in black. QWK is omitted because it is not applicable to individual cases.

**CR Loss.** To evaluate the impact of each component in our loss function, we conduct an ablation study on metric function used in Equation (6). The results shown in Table 4 indicate the proposed CR distance can not only perform better in MCRE and AMCRE which are relevant with clinical risk, but also in traditional ordinal regression metrics.

**Stratified results.** Similarly, Table 5 and Table 6 report the stratified results across individual grades. It can be observed that our method consistently outperforms alternative data utilization strategies and metric functions across the majority of grades.

Table 3: Ablation study on CFR module.

| Data Type | | QWK↑ | MAE↓ | AMAE↓ | MCRE↓ | AMCRE↓ |
|---|---|---|---|---|---|---|
| 3D SCPR | CCTA | | | | | |
| ✓ | | 0.692 | **0.473** | 0.557 | 1.453 | 1.300 |
| | ✓ | 0.694 | 0.506 | 0.568 | 1.335 | 1.215 |
| ✓ | ✓ | **0.712** | 0.480 | **0.533** | **1.296** | **1.186** |

Table 4: Ablation study on CR Loss.

| Metric Function | | | QWK↑ | MAE↓ | AMAE↓ | MCRE↓ | AMCRE↓ |
|---|---|---|---|---|---|---|---|
| AD | SD | CRD | | | | | |
| ✓ | | | 0.689 | **0.475** | 0.569 | 1.430 | 1.277 |
| | ✓ | | 0.689 | 0.511 | 0.564 | 1.445 | 1.249 |
| | | ✓ | **0.712** | 0.480 | **0.533** | **1.296** | **1.186** |

Table 5: Stratified results of the ablation study on CFR module.

| Data Type | | MAE↓ | | | | | MRCE↓ | | | | |
|---|---|---|---|---|---|---|---|---|---|---|---|
| 3D SCPR | CCTA | 1 | 2 | 3 | 4 | 5 | 1 | 2 | 3 | 4 | 5 |
| ✓ | | **0.234** | 0.684 | 0.496 | 0.672 | 0.700 | 1.052 | 2.040 | 1.200 | 1.507 | 0.700 |
| | ✓ | 0.333 | 0.669 | 0.496 | **0.642** | 0.700 | 0.854 | 1.974 | 1.174 | **1.373** | 0.700 |
| ✓ | ✓ | 0.279 | **0.654** | **0.478** | **0.642** | **0.600** | **0.838** | **1.923** | **1.130** | 1.478 | **0.600** |

Table 6: Stratified results of the ablation study on CR Loss.

| Metric Function | | | MAE↓ | | | | | MRCE↓ | | | | |
|---|---|---|---|---|---|---|---|---|---|---|---|---|
| AD | SD | CRD | 1 | 2 | 3 | 4 | 5 | 1 | 2 | 3 | 4 | 5 |
| ✓ | | | **0.247** | 0.684 | 0.487 | 0.627 | 0.800 | 1.018 | 2.070 | 1.139 | 1.358 | 0.800 |
| | ✓ | | 0.333 | 0.691 | 0.513 | **0.582** | 0.700 | 1.053 | 2.092 | 1.191 | **1.209** | 0.700 |
| | | ✓ | 0.279 | **0.654** | **0.478** | 0.642 | **0.600** | **0.838** | **1.923** | **1.130** | 1.478 | **0.600** |

## 5. Conclusion

In this paper, we propose a novel deep learning-based approach to achieve multi-level stenosis grading for the first time, as far as we know. To address the challenge of predicting stenosis grade accurately in the presence of abnormal distortions in 3D SCPR images, we introduce a novel CFR module that allows for a comprehensive stenosis analysis from both 3D SCPR and CCTA images. Furthermore, we incorporate the clinical relevance of stenosis grading by introducing CR Loss, which encodes the clinical risk boundary into neural network training. Through experiments conducted on a private dataset, we demonstrate that our method outperforms other approaches significantly in both traditional and clinical relevance metrics.

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
