# OpenReview forum: "Clinical Risk-Aware Multi-Level Grading for Coronary Artery Stenosis through Curved Feature Reconstruction"
_MIDL.io/2026/Conference — MIDL 2026 Poster_

### Official Review · Reviewer_UpUn · 2026-01-05

**Confidence:** 4
**Preliminary Rating:** 3
**Final Rating:** 5

**Summary:**

The paper proposes a method to classify coronary artery stenosis into five stenosis grades, positioned as a trade-off between binary classification and regression-based approaches. The method leverages both projected planar representations and the original CCTA images and introduces a novel loss function formulation.

**Strengths:**

The paper addresses a clinically relevant problem, and the presented results are promising. In addition, the methodology appears novel and offers a new perspective compared to existing approaches for stenosis grading.

**Weaknesses:**

The paper’s framing raises several clarity and consistency concerns. First, the authors introduce 3D Straightened Curved Planar Reformation (3D SCPR) images as a completely different imaging modality from CCTA and critique existing methods for relying on this modality. However, 3D SCPR images are derived from CCTA, making this distinction and criticism potentially misleading.
Second, the motivation for five-grade stenosis classification is not fully convincing. The authors state that treating different stenosis grades equally does not meet the demands of clinical practice, yet earlier they indicate that clinical significance primarily lies between grades 1–2 and 3–5. This inconsistency weakens the justification for a five-class formulation. Additionally, the rationale for preferring classification over regression is not sufficiently developed. A comparison against regression-based approaches (followed by discretization into five grades) would strengthen this argument.
Finally, in the qualitative evaluation (Figure 3), it is unclear which methods are being compared, as the figure lacks citations and the method names differ from those introduced in the previous sections, making the comparison difficult to interpret.

**Detailed Comments:**

n/a

**Justification Of Final Rating:**

The underlying idea of the paper is interesting and relevant. I thank the authors for the modifications and clarifications provided in the revision, which address the key issues raised in my initial review. The presentation is now clearer, and the method and its relevance are more convincingly conveyed.

**Justification Of The Preliminary Rating:**

I believe that several key aspects need to be addressed for the paper to be fully convincing. While the underlying idea is interesting, important details are not clearly conveyed, which ultimately affects the understanding of the method and its relevance.

**Questions To Address In The Rebuttal:**

-The paper would benefit from a clearer justification for the five-grade classification scheme, and/or from a comparison with regression-based approaches, for example by regressing stenosis severity and discretizing the predictions into five grades.

-In addition, a more detailed explanation of 3D SCPR images would improve clarity, including how they are derived from CCTA data and what types of artifacts or limitations this representation may introduce.

---

> ### Author Response · Authors · 2026-01-25
> **Response to Reviewers**
>
> >**【W1】The motivation for five-grade stenosis classification is not fully convincing ... they indicate that clinical significance primarily lies between grades 1–2 and 3–5.**
>
> **Response:**
>
> We appreciate the reviewer’s query regarding the clinical rationale. We would like to clarify that our 5-grade classification scheme is strictly aligned with the **CAD-RADS™  2.0** [1].
> * **Standardization Requirement:** As CAD-RADS™ 2.0 represents the **preeminent global clinical standard** for CCTA-based coronary artery disease diagnosis—jointly established by the Society of Cardiovascular Computed Tomography and other leading organizations—it provides a **standardized framework** for grading stenosis from 1 to 5. Consequently, **to better support physicians** in generating standardized clinical reports, we believe it is essential for our model to output these specific grades.
>
> * **Justification for CR Loss:** The reviewer correctly noted the distinction between grades 1-2 and 3-5. According to **Table 4 of [1]**, **grades 1-2** typically require "non cardiac investigation", while **grades 3-5** trigger "functional assessment or invasive coronary angiography (ICA)".
>     * This implies that while we classify 5 grades, the **penalty for misclassification across the boundary** (e.g., confusing Grade 2 with Grade 3) should be significantly higher than misclassification within a group (e.g., confusing Grade 3 with Grade 4).
>     * This clinical reality is precisely the motivation behind our CR Loss. It reconciles the need for 5-grade reporting with the uneven clinical risks associated with different error types.
>
> >**【W2】The rationale for preferring classification over regression is not sufficiently developed. A comparison against regression-based approaches ... would strengthen this argument.**
>
> **Response:**
>
> We thank the reviewer for this constructive suggestion. We fully agree that providing an empirical comparison in our specific context is necessary to strengthen the paper. So we add a **Continuous Label Regression** approach in the experiment. The specific details can be found in **Section 4.2 and Table 1 of the revised manuscript**.
>
> >**【W3】The authors introduce 3D SCPR images as a completely different imaging modality... A more detailed explanation of 3D SCPR images would improve clarity.**
>
> **Response:**
>
> We apologize for any confusion caused by our terminology. We have **added Section 3.1** to provide a detailed description of the generation methodology of 3D SCPR images in the revised manuscript. For further information on the foundational SCPR technique, please also refer to [2, 3]. To further improve clarity and facilitate future research, we will **release the source code** for generating 3D SCPR images upon acceptance.
>
> **Clarification:**
>
> * **Derivation:** 3D SCPR is generated by sampling the original CCTA image along the vessel centerline. Crucially, this representation incorporates prior knowledge (the centerline sequence) which the raw CCTA images lacks. Moreover, it is common practice to use 3D SCPR images, and they have been extensively used in many related works [4, 5].
>
> * **Complementarity:** We utilize both views because they offer complementary advantages:
>     * 3D SCPR: By incorporating the vessel centerline as prior information, 3D SCPR images enable the algorithm to perceive the vessels in a straightened state, thereby educing geometric complexity and simplifying the identification of stenosis. However, due to the inherent torsion of 3D centerlines, local planar grids tend to rotate inconsistently at different positions. This leads to rotational misalignment, where the radial orientation of points fails to align, causing anatomical structures to appear twisted or spiraled in the final 3D SCPR images. This is a systemic issue inherent to the design.
>     * Original CCTA: Preserves the original spatial context and density information without deformation.
>     * Our method leverages both inputs to maximize feature extraction robustness.
>
> >**【W4】 In the qualitative evaluation (Figure 3), it is unclear ... making the comparison difficult to interpret.**
>
> **Response:**
>
> We thank the reviewer for pointing out this lack of clarity. The figure is intended to validate the effectiveness of our CFR module by comparing different data utilization strategies: "CCTA-only", "3D SCPR-only", and "Ours (CCTA + 3D SCPR)". We have **revised Figure 3 and its caption in the revised manuscript**.
>
> **References:**
>
> [1] Cury et al. Cad-rads™ 2.0–2022 coronary artery disease-reporting and data system. Cardiovascular Imaging, 2022.
>
> [2] Dupej et al. Blood Vessel Visualization on CT Data. 2012.
>
> [3] Felkel et al. CPR-curved planar reformation. IEEE, 2002
>
> [4] Zhang et al. Coronary r-cnn: Vessel-wise method for coronary artery lesion detection and analysis in coronary ct angiography. MICCAI 2022.
>
> [5] Ma et al. Transformer network for significant stenosis detection in ccta of coronary arteries. MICCAI 2021.

---

> ### Comment · Area_Chair_cBZa · 2026-02-01
> **Final rating**
>
> Dear reviewer,
>
> Could you please provide your final rating for this submission?
>
> Thank you!

---

### Official Review · Reviewer_7dbP · 2026-01-07

**Confidence:** 5
**Preliminary Rating:** 4
**Final Rating:** 5

**Summary:**

The paper tackles the problem of Coronary Artery Stenosis prediction from Cardiac CT Angiography volumes. The key significances of this paper are two-fold.

First, it introduces novel metrics and losses to assess prediction error of a clinical parameter (here, stenosis grading) whilst taking into account the fact that such clinical parameters have ranges of values which determine treatment choice - therefore, an algorithm should be penalized more if predicting in one range when the true value is in another, compared to when the algorithm mispredicts but it is still within the clinical range of that treatment choice.

Second, it introduces a machine learning architecture which allows combining centerline reformed view CT (SCPR) and raw CT volume (CCTA) information and aligning their feature spaces, with the goal of avoiding loss of texture information due to distortion of the original CT while still retaining the relationship between centerline coordinates and the underlying stenosis assessment.

The paper then shows with extensive qualitative and quantitative evaluation that the proposed algorithm outperforms baseline and CCTA-only and SCPR-only algorithms, in both traditional and the newly proposed metrics.

The paper is highly significant to the medical imaging and computed-aided diagnosis device field, as it tackles the problem of bringing clinical importance to the development of machine learning-based devices, successfully showing a new approach that may also be applied to other domains due to its inherently general nature.

**Strengths:**

The paper has several clear strengths.

1) It introduces clinically informed metrics and loss functions that reflect how stenosis grades are used in real treatment decisions. This is valuable because it penalizes clinically meaningful errors more than minor mis-predictions within the same treatment range. The idea is general and can be applied to many other medical prediction tasks with ordinal or range-based outcomes.

2) The proposed architecture combines SCPR and raw CCTA data while aligning their feature spaces relative to the annotated centerline effectively leverages complementary information, preserving anatomical relationships from centerline views while avoiding texture loss from CT distortion.

3) The work provides thorough qualitative and quantitative evaluation, showing consistent improvements over SCPR-only, CCTA-only, and baseline methods across both standard and proposed metrics.

4) The writing is clear, well structured, and grounded in relevant prior work.

**Weaknesses:**

The paper has no significant weaknesses.

However, it would help if more example images such as the ones in Figure 3 were shown - especially the failure cares.

Additionally, since the dataset used was imbalanced in terms of stenosis grades, a stratification of the results is would be helpful if combined with appropriate statistical analysis to confirm the assertions of superior performance of the proposed approaches, not only in the full test group but also in different stenosis subgroups.

Lastly, loss curves comparing the different compared models would be great adds to the work.

**Detailed Comments:**

It would be interesting to also compare against works that try to segment the lumen and wall directly, instead of simply predicting the stenosis degree directly.

**Justification Of Final Rating:**

The authors concisely addressed all my concerns. Response to my comments were also satisfying and reasonable. Aligning views from multiple modality and within single modality is extremely relevant for the field.

**Justification Of The Preliminary Rating:**

I give this paper a weak accept because it presents a meaningful, well-written, and well-motivated contribution. The proposed clinically informed metrics and loss functions, together with the multimodal SCPR–CCTA architecture, address an important gap between algorithmic performance and clinical relevance. The approach is novel, generalizable, and supported by thorough experimental results that consistently outperform strong baselines.

However, the paper would benefit from additional qualitative examples (especially failure cases), stratified analysis to account for dataset imbalance, and more training diagnostics such as loss curves. These limitations are relatively minor and do not undermine the core contributions, but they slightly reduce the paper’s overall completeness.

Overall, the strengths clearly outweigh the weaknesses, supporting a weak accept rating.

**Questions To Address In The Rebuttal:**

These points (ordered by importance from high to low), if addressed, would help get a better feeling of the significance of the outcome asserted by this paper.

1) Please provide figures like Figure 3 but from a random sample of cases to support the assertion of superior performance of the proposed algorithm. In this new figure, also show the exact numbers for the different metrics at the individual subject level.
2) Please provide stratified results for tables 1 and 2, with rows containing only cases with stenosis grades 1, 2, 3, 4, or 5 - to show wether the newly proposed approach leads to bias towards certain stenosis grades. Also show the total number of cases used in each row.
3) Please also show loss curves for all algorithms trained

---

> ### Author Response · Authors · 2026-01-25
> **Response to Reviewers**
>
> >**【Q1】Please provide figures like Figure 3 but from a random sample of cases... In this new figure, also show the exact numbers for the different metrics at the individual subject level.**
>
> **Response:**
>
> We thank the reviewer for this valuable suggestion. Visualizing a broader range of samples is indeed essential to provide a transparent view of the model's performance.
>
> **Action:**
>
> We have updated **Figure 3 in the revised manuscript**.
>
> * We **randomly selected 10 examples** from the test set to ensure a representative display of the model's capabilities.
> * For each case, we have explicitly annotated **the exact values of the metrics** at the individual subject level. We believe this updated visualization offers a more comprehensive qualitative assessment of our proposed method.
>
> >**【Q2】Please provide stratified results for tables 1 and 2, with rows containing only cases with stenosis grades 1, 2, 3, 4, or 5... Also show the total number of cases used in each row.**
>
> **Response:**
>
> This is an excellent suggestion. Given the class imbalance inherent in medical datasets, we agree that verifying the performance across different subgroups is crucial to ensure the model does not exhibit bias towards specific stenosis grades. We have **revised all tables** to include detailed **stratified results**. The specific details can be found in **Section 4.2, 4.3 and Table 2, 5, 6 in the revised manuscript**.
>
> >**【Q3】Please also show loss curves for all algorithms trained.**
>
> **Response:**
>
> We appreciate the reviewer’s suggestions. We carefully considered adding a comparison of loss curves.
>
> **Explanation:**
>
> However, we would like to respectfully clarify why we decided not to include a direct comparison plot of the loss curves in the final manuscript. The comparison methods used in our study employ different loss functions with significantly **different magnitudes**. For instance, our **CR Loss** incorporates a specific **loss weights vector**, whereas the loss in **Multi-class Classification** is **unweighted**. Furthermore, the **Cost-sensitive Loss** introduces an additional **cost-sensitive term** compared to Multi-class Classification. Meanwhile, the **Continuous Label Regression** task employs **MSE loss**, which differs from the aforementioned classification-based losses. Plotting them on the same graph would result in inconsistent baselines, making a direct visual comparison mathematically invalid and potentially misleading. Instead, we believe the rigorous quantitative comparisons (in Table 1) provide the most accurate assessment of the superiority of our approach. We hope this explanation is acceptable.

---

### Official Review · Reviewer_3SMu · 2026-01-11

**Confidence:** 5
**Preliminary Rating:** 5
**Final Rating:** 5

**Summary:**

This paper introduces a deep learning framework for multi-level grading of coronary artery stenosis that integrates both CCTA and 3D SCPR images through a Curved Feature Reconstruction (CFR) module, which aligns vessel features using curve priors to overcome distortions. It proposes a Clinical Risk-Aware (CR) Loss, which penalizes misclassifications across critical risk boundaries between mild and moderate-to-severe stenosis grades. Experiments show that the method significantly outperforms existing baselines, particularly in metrics designed to capture clinical risk relevance (MCRE and AMCRE). Ablation studies confirm that both CFR and CR Loss are essential, and qualitative visualizations demonstrate that combining modalities improves localization of stenosis regions. The significance lies in aligning algorithmic grading with clinical decision-making, offering a more reliable and risk-sensitive tool for CAD diagnosis.

**Strengths:**

The paper introduces the Curved Feature Reconstruction (CFR) module and the Clinical Risk-Aware (CR) Loss, both of which directly address limitations in prior work. CFR effectively fuses CCTA and 3D SCPR features to overcome distortions, while CR Loss explicitly encodes clinically meaningful risk boundaries into training. These innovations make the method more aligned with real-world diagnostic practice. The approach is well-formulated, with clear mathematical definitions and a structured pipeline. The inclusion of both quantitative and qualitative experiments, along with ablation studies, demonstrates careful validation of each component.

**Weaknesses:**

The experiments are conducted on a single in-house dataset, which restricts the generalizability of the findings. Without external validation on publicly available datasets or multi-center cohorts, it is difficult to assess whether the method would perform consistently across diverse patient populations and imaging protocols.
While the Clinical Risk-Aware (CR) Loss is a novel idea, the justification for the specific penalty design (squared error plus offset across the risk boundary) is somewhat heuristic. A more thorough comparison with established ordinal regression or cost-sensitive learning frameworks (e.g., those used in medical risk stratification) would strengthen the claim of superiority.

**Detailed Comments:**

Provide more information about the in-house dataset, including patient demographics, imaging protocols, and sample size distribution across stenosis grades. This will help readers assess the robustness and generalizability of the results.

**Justification Of Final Rating:**

It combines methodological novelty with clear clinical relevance. The introduction of the Curved Feature Reconstruction (CFR) module directly addresses distortions in 3D SCPR images by fusing them with undistorted CCTA features, which is a technically sound and clinically motivated solution. The Clinical Risk-Aware (CR) Loss is another valuable contribution, as it explicitly encodes the risk boundaries between mild and moderate-to-severe stenosis grades, aligning the algorithm’s training objectives with real-world diagnostic practice.

**Justification Of The Preliminary Rating:**

It combines methodological novelty with clear clinical relevance. The introduction of the Curved Feature Reconstruction (CFR) module directly addresses distortions in 3D SCPR images by fusing them with undistorted CCTA features, which is a technically sound and clinically motivated solution. The Clinical Risk-Aware (CR) Loss is another valuable contribution, as it explicitly encodes the risk boundaries between mild and moderate-to-severe stenosis grades, aligning the algorithm’s training objectives with real-world diagnostic practice.

**Questions To Address In The Rebuttal:**

.

---

> ### Author Response · Authors · 2026-01-25
> **Response to Reviewers**
>
> >**【W1】The experiments are conducted on a single in-house dataset, which restricts the generalizability of the findings. Without external validation on publicly available datasets or multi-center cohorts, it is difficult to assess whether the method would perform consistently across diverse patient populations and imaging protocols.**
>
> **Response:**
>
> We fully agree with the reviewer that multi-center validation on public datasets is the gold standard for evaluating generalizability. However, we would like to respectfully clarify the specific challenges in the field of CCTA-based stenosis analysis that currently limit the availability of such data:
>
> * **Complexity of Annotation:** Unlike 2D medical images, CCTA is high-dimensional **3D volumetric data**. Accurate annotation of stenosis requires experienced radiologists to review the scan **slice-by-slice** to determine the degree of narrowing. This process is extremely labor-intensive, time-consuming, and costly, making large-scale data collection significantly harder than in other medical imaging domains.
> * **Lack of Public Benchmarks:** To the best of our knowledge, there are currently **no publicly available CCTA datasets with fine-grained stenosis grade annotations**. Consequently, reliance on in-house datasets is the standard practice in this specific research area, as seen in recent related works [1, 2].
> * **Privacy and Regulations:** Strict data privacy regulations further complicate the sharing of such sensitive 3D patient data across institutions.
>
> Despite these objective constraints, we acknowledge the importance of validation. **In our future work, we are committed to continuing our data collection efforts** and aim to establish collaborations to verify the proposed algorithm on broader cohorts.
>
> >**【W2】While the Clinical Risk-Aware (CR) Loss is a novel idea, the justification for the specific penalty design... is somewhat heuristic. A more thorough comparison with established ordinal regression or cost-sensitive learning frameworks... would strengthen the claim of superiority.**
>
> **Response:**
>
> We appreciate this excellent suggestion. Comparing our method with established frameworks is indeed crucial to justify the design of our CR Loss. Following your recommendation, we have added comparison with **a cost-sensitive classification method** (referenced from [3]) to validate the effectiveness of our approach. The specific details can be found in **Section 4.2 and Table 1 of the revised manuscript**.
>
> **References:**
>
> [1] Zhang et al. Coronary r-cnn: Vessel-wise method for coronary artery lesion detection and analysis in coronary ct angiography. MICCAI 2022.
>
> [2] Ma et al. Transformer network for significant stenosis detection in ccta of coronary arteries. MICCAI 2021.
>
> [3] Galdran et al. Cost-sensitive regularization for diabetic retinopathy grading from eye fundus images. MICCAI 2020.

---

### Author Rebuttal · Authors · 2026-01-25

**Rebuttal:**

Dear Editor and Reviewers,

We would like to express our sincere appreciation to the reviewers for their insightful comments and constructive suggestions. These contributions have been invaluable in enhancing the technical depth and clarity of our manuscript.

We have carefully addressed all the concerns raised and have submitted a revised version of our manuscript. All major revisions are highlighted for easy identification.

In response to the collective feedback, we have implemented several significant updates to the manuscript:

* **Detailed Methodology (New Section 3.1):** We add Section 3.1, which provides a comprehensive description of the 3D SCPR image generation process..

* **Expanded Experimental Results (Sections 4.2 & 4.3):** We add additional experiments comparing **Cost-sensitive Classification** and **Continuous Label Regression** with our proposed method. Meanwhile, we introduce **stratified results** for all experiments.

* **Visual Improvements (Figure 3):** We update Figure 3, the new figure showcases 10 randomly selected samples and ensures a representative display of the model's capabilities.

**Supporting Material:**

/attachment/26d424cbefbc81584a03466ce536c00bace4ebe1.pdf

---

### Meta-Review · Area_Chair_cBZa · 2026-02-07

**Recommendation:** Accept (Poster)
**Confidence:** 5

**Metareview:**

All reviewers were positive about the two contributions of this paper. First, the combination of axial and reformatted CCTA patches into a model for coronary stenosis detection. Second, the use of a new type of ordinal loss for multiclass classification to emphasize the clinical distinction between grades 1-2 and 3-4-5 coronary stenosis. My recommendation is to accept this paper.

However, I'd strongly urge the authors to reconsider reviewer UpUn's comments on the distinction between SCPR and CCTA. The authors present these two as distinct image modalities, but in fact, what they call 'CCTA' is just an axial view of the same underlying CCTA image that provides the SCPR. A better way to refer to these two inputs would thus be 'axial CCTA' and 'SCPR CCTA'.

Moreover, I'd recommend that the authors conduct a thorough additional proofreading of the manuscript, as the style can be substantially improved. In addition, the abstract can be improved for improved clarity and to better reflect the manuscript's content. E.g., it's unclear what SCPR stands for, and what the authors mean by 'multi-level grading for coronary artery stenosis'.

Finally, the Introduction erroneously states that clinicians use (axial) CCTA for stenosis grading. In practice, clinicians use reformatted images, which is exactly why previous DL methods have treated such reformatted images (here referred to as 'SCPR') as inputs.

---

### Decision · Program_Chairs · 2026-02-13

Accept (Poster)